# REPOBENCH: BENCHMARKING REPOSITORY-LEVEL CODE AUTO-COMPLETION SYSTEMS

**Tianyang Liu    Canwen Xu    Julian McAuley**
University of California San Diego
{til040, cxu, jmcauley}@ucsd.edu

## ABSTRACT

Large Language Models (LLMs) have greatly advanced code auto-completion systems, with a potential for substantial productivity enhancements for developers. However, current benchmarks mainly focus on single-file tasks, leaving an assessment gap for more complex, real-world, multi-file programming scenarios. To fill this gap, we introduce RepoBench, a new benchmark specifically designed for evaluating repository-level code auto-completion systems. RepoBench supports both Python and Java and consists of three interconnected evaluation tasks: RepoBench-R (Retrieval), RepoBench-C (Code Completion), and RepoBench-P (Pipeline). Each task respectively measures the system's ability to retrieve the most relevant code snippets from other files as cross-file context, predict the next line of code with cross-file and in-file context, and handle complex tasks that require a combination of both retrieval and next-line prediction. RepoBench aims to facilitate a more complete comparison of performance and encouraging continuous improvement in auto-completion systems. RepoBench is actively maintained with the latest code, serving as a live benchmark publicly available at https://github.com/Leolty/repobench.

## 1 INTRODUCTION

Large language models (LLMs; Brown et al., 2020; Chowdhery et al., 2022; Touvron et al., 2023; OpenAI, 2023; Chiang et al., 2023; Xu et al., 2023) have been instrumental in paving new avenues for innovative applications across diverse domains, with programming being a notably attractive and promising domain (Chen et al., 2021; van Dam et al., 2023; Austin et al., 2021; Wang et al., 2023). In particular, the rise and application of code auto-completion systems like GitHub's Copilot [1], driven by OpenAI's Codex (Chen et al., 2021), have the potential to substantially changed the manner in which we interact with code. These changes facilitate coding for beginners and improve efficiency of the coding process for experienced developers.

A variety of code auto-completion models (Chen et al., 2021; Guo et al., 2022; Fried et al., 2022; Nijkamp et al., 2022; Li et al., 2023; Allal et al., 2023; Guo et al., 2023; 2024) have emerged in recent years, each boasting unique capabilities and performance characteristics. This emergence of models emphasizes the increasing importance of AI in the realm of programming, leading a more diversified and competitive landscape. However, current evaluation datasets and benchmarks (Lu et al., 2021; Raychev et al., 2016a; Allamanis & Sutton, 2013) predominantly focus on completion tasks within the scope of a single file. This focus fails to reflect the complexity and intricacies of real-world programming scenarios, where developers frequently work on multi-file projects, often navigating through and understanding code spanning several repositories.

Recognizing the need for a more comprehensive evaluation, we introduce RepoBench, a new benchmark for evaluating the effectiveness of repository-level code auto-completion systems. Specifically, RepoBench offers three distinct evaluation sub-tasks, each emphasizing a unique aspect of a fully functioning code auto-completion system: (1) **The Retrieval Task (RepoBench-R)**, which tests the system's ability to retrieve the most relevant code snippets, thereby providing the necessary context for the prediction of the next line of code. (2) **The Code Completion Task (RepoBench-C)**, where

---

[1] https://github.com/features/copilot

the task is to predict the next line of code given a pre-defined context. The context can involve content from different files (cross-file context) and within the file (in-file context) with a moderate length setting that can fit most models. (3) **The End-to-End Pipeline Task (RepoBench-P)**, which is designed to simulate the complete process of a code auto-completion system like GitHub Copilot - first retrieving relevant snippets and then completing the code by predicting the next line. In this scenario, the system may encounter a large set of potential snippets for retrieval, resulting in longer and broader contexts, which leads to the need for the system to optimize the efficient selection of numerous candidates to facilitate code completion while ensuring that the extensive context remains within the system's processing capabilities.

To summarize, the primary contributions of our work are as follows:

- We present RepoBench, a benchmark tailored for evaluating repository-level code auto-completion systems. This benchmark comprises three interconnected tasks: RepoBench-R for code retrieval, RepoBench-C for code completion, and RepoBench-P, which integrates both aspects to reflect the entire workflow of an auto-completion system, offering a balanced assessment.

- We conduct a series of experiments on RepoBench, analyzing the efficacy of various retrieval methods and code completion models of different magnitudes, and the assessment of their combined performance in a full pipeline, providing some insights for future research and development. Our results underscore the significance of code models that can manage extended contexts and maintain generalizability in real-world coding environments.

## 2    RELATED WORK

**LLMs for Code Completion**  Code completion, also referred to as auto-completion or intelligent code completion, is an essential feature provided by many modern Integrated Development Environments (IDEs) and code editors. It aids programmers in writing code more efficiently by predicting and automatically completing the next line or multiple next lines. The inception of Language Models (LMs) in code completion can be traced back to the usage of n-gram based LMs (Tu et al., 2014; Hindle et al., 2016), RNN models (White et al., 2015), and probabilistic grammar-models (Bielik et al., 2016; Raychev et al., 2016b; Hellendoorn & Devanbu, 2017), which laid the foundation for the subsequent introduction of more advanced LMs in this field. With the advent of transformer-based models (Vaswani et al., 2017; Devlin et al., 2019; Radford et al., 2018; 2019; Brown et al., 2020), decoder-only models trained on large-scale code datasets have been proposed to foster the advancements in code completion. For instance, GPT-C (Svyatkovskiy et al., 2020a) and CodeGPT (Lu et al., 2021) following the underlying architecture of GPT-style models are pre-trained on vast amounts of code. UniXCoder (Guo et al., 2022) and CugLM (Liu et al., 2020) incorporates multi-task learning strategies, and leverages code structures to enhance pretraining. More recent LLMs, including Codex (Chen et al., 2021), PolyCoder (Xu et al., 2022), CodeGen (Nijkamp et al., 2022), In-Coder (Fried et al., 2022), CodeGeeX (Zheng et al., 2023), SantaCoder (Allal et al., 2023), Star-Coder (Li et al., 2023; Lozhkov et al., 2024), LongCoder (Guo et al., 2023), CodeLlama (Rozière et al., 2024) and DeepSeekCoder (Guo et al., 2024) employ billions of parameters and excel in code generation tasks, benefiting from large-scale, high-quality code corpora. The scope of code completion has expanded with works like RLPG (Shrivastava et al., 2022), CoCoMIC (Ding et al., 2022), and RepoCoder (Zhang et al., 2023), emphasizing the integration of in-file and cross-file contexts and the importance of specialized benchmarks for evaluating repository-level code autocompletion systems.

**Code Completion Datasets**  The task of code completion serves as a foundation for programming language models and plays a pivotal role in intelligent code completion systems. While public benchmarks like CodeXGLUE (Lu et al., 2021) with datasets *PY150* (Raychev et al., 2016a) and *Github Java Corpus* (Allamanis & Sutton, 2013) play a key role in evaluating models within single-file contexts, they may not fully encapsulate the intricacies of real-world coding projects which often entail cross-file interactions. To address this, Ding et al. (2022) proposed CoCoMIC, a model for cross-file completion and a code completion dataset with retrieved cross-file context. Different from the CoCoMIC data, our benchmark extends beyond code completion and includes evaluation of retrieval and pipeline construction, thus can better capture the complexity of such cross-file code completion systems. RepoEval by Zhang et al. (2023) serves as a project-oriented benchmark,

focusing on 16 selected Python repositories to simulate real-world coding environments. However, its limitation arises from being integrated into the training data of StarCoder. RepoBench not only spans a wider range of repositories across Python and Java, but also offers a segmented evaluation into retrieval, completion, and end-to-end tasks.

Transitioning from file-based to repository-level code completion not only offers a more realistic representation of practical coding scenarios but also serves as a platform for evaluating the transfer learning capabilities of language models, as most models are not initially pre-trained with cross-file contexts included. This shift also introduces the challenge of handling longer prompts, a situation less common in single-file contexts, and a known limitation of many Transformer-based models. Recent research on long-range transformers (Zaheer et al., 2020) has shown promise in handling long sequences, with notable contributions from initial works like LongFormer (Beltagy et al., 2020) and Reformer (Kitaev et al., 2020), as well as more recent advancements like CoLT5 (Ainslie et al., 2023), UnlimiFormer (Bertsch et al., 2023), and Claude-100k (PBC, 2023), which has demonstrated their potential in effectively processing and generating code with much more cross-file context included.

## 3 THE REPOBENCH DATASET

RepoBench is a live benchmark for auto-code completion, with a commitment to continuously incorporate the latest data for model evaluation. This paper introduces the construction and findings of RepoBench's inaugural iteration (v1.0).

### 3.1 DATA SOURCES

**Github-Code Dataset:** The first source of RepoBench is the `github-code` dataset[2], which consists of a vast collection of code files sourced from GitHub repositories under open-source licenses with a data cutoff date of *March 16, 2022*. Specifically, we aggregate files based on their repository name as the github-code dataset is originally stored at the file-level. Given that the code in this dataset has been widely utilized for training various models (Li et al., 2023; Nijkamp et al., 2022), we primarily use this dataset for constructing our training data. The use of this data for training specifically addresses the adoption of patterns that concatenate cross-file context and in-file context for next-line prediction. Fine-tuning on this dataset is optional, as sufficiently robust models may already exhibit this generalizability.

**Newly Crawled GitHub Data:** To mitigate impacts regarding data leakage and memorization, we augment the dataset by incorporating the most recent, non-forked GitHub repositories that are permitted under their respective licenses. Specifically, we use GitHub's official API to crawl Python and Java repositories created after *February 9, 2023*, which aligns with the newest knowledge cutoff date of The Stack (Kocetkov et al., 2022), and before *August 3, 2023*. This newly-crawled data serves exclusively as our test set for evaluation.

**Continuous Updates:** In response to the rapid advancement of Code LLMs and their training datasets, RepoBench is committed to a regimen of continuous updates, to ensure that RepoBench keeps pace with the latest developments and avoids potential data leakage, which could compromise the integrity of model evaluations. As of this writing, RepoBench v1.1 is already available. Detailed discussions on RepoBench v1.1 can be found Appendix C.

### 3.2 DATA PROCESSING

The data processing procedure for this study involves multiple steps. For the training data sourced from `github-code`, repositories with a number of Python or Java files between 32 and 128 are selected. This range is chosen to ensure an adequate cross-file dependency while avoiding excessive complexity and keeping the data volume within a reasonable range. While for the newly crawled test data, we do not set file number constraints to ensure a thorough evaluation. To identify cross-file dependencies and their usage, we use tree-sitter[3] to parse each file. This parsing is primarily directed at import statements, enabling us to identify all cross-file modules and the lines utilizing these

---

[2]`https://huggingface.co/datasets/codeparrot/github-code`
[3]`https://tree-sitter.github.io/tree-sitter/`

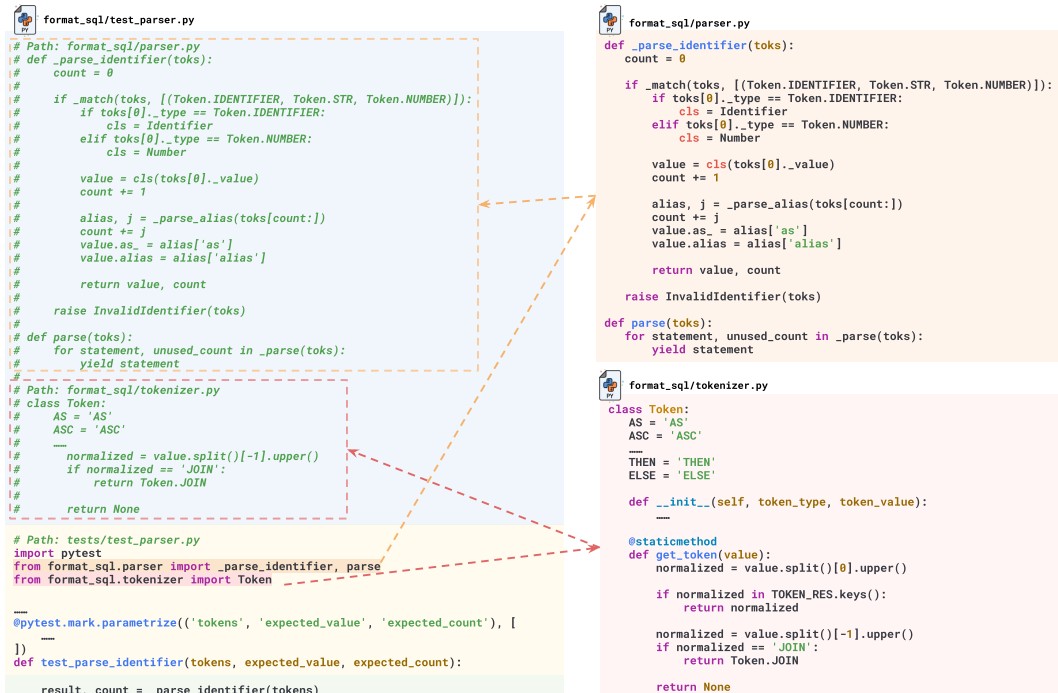

Figure 1: Construction of a prompt for repository-level cross-file code completion. The commented cross-file context (path + snippet), parsed from import statements using `tree-sitter`, is concatenated with the in-file context (path + import statements + preceding lines), which cropped to a maximum of 30 lines in RepoBench to form the input prompt, with the objective is to predict the next line . Note that for clarity, certain lines of code are omitted in this figure, which is an abbreviated and simplified version derived from a real example. Refer to Appendix A for a detailed ablation study on prompt construction.

modules (termed cross-file lines). Further, we track the corresponding code snippets that define these imported modules.

After processing the data, our dataset comprises 10,345 Python and 14,956 Java historical repositories, serving as training data and are available for optional fine-tuning. Additionally, we have 1,075 Python and 594 Java new repositories from GitHub designated as test data for evaluation.

## 3.3 TASK CONSTRUCTION

**Task Settings** To effectively evaluate next-line prediction in auto-completion systems, we define three settings:

- **Cross-File-First (XF-F):** This is the most challenging setting, where we mask the first appearance of a cross-file line within a file. In this setting, there is no prior usage of the module in the in-file context to aid the prediction, thereby requiring the system to handle long-range cross-file context for better accuracy.

- **Cross-File-Random (XF-R):** In this setting, we mask a random and non-first occurrence of a cross-file line. Unlike the XF-F setting, the prior in-file usage of the module may serve as a hint for the prediction.

- **In-File (IF):** In this setting, we mask an in-file line that does not involve any cross-file modules. This setting serves as a robustness test to ensure that the incorporation of cross-file context does not greatly affect the accuracy of predictions.

Note that RepoBench-R (Retrieval) is designed with only XF-F and XF-R settings, as IF does not involve retrieval and thus cannot be evaluated in this task, while both RepoBench-C (Code Completion) and RepoBench-P (Pipeline) involve all three settings: XF-F, XF-R, and IF.

Table 1: Overview of the test data in RepoBench for Python and Java across three tasks: RepoBench-R, RepoBench-C and RepoBench-P, including details on the number of data points for different settings (XF-F, XF-R, IF), as well as the mean number of candidates and tokens in each subset.

| Lang. | Task | Subset | XF-F | XF-R | IF | Mean Candidates | Mean Tokens |
|---|---|---|---|---|---|---|---|
| Python | RepoBench-R | Easy | 12,000 | 6,000 | - | 6.7 | - |
| | | Hard | 12,000 | 6,000 | - | 17.8 | - |
| | RepoBench-C | 2k | 12,000 | 5,000 | 7,000 | - | 1,035 |
| | | 8k | 18,000 | 7,500 | 10,500 | - | 3,967 |
| | RepoBench-P | | 10,867 | 4,652 | 6,399 | 24 | 44,028 |
| Java | RepoBench-R | Easy | 12,000 | 6,000 | - | 6.8 | - |
| | | Hard | 12,000 | 6,000 | - | 25.5 | - |
| | RepoBench-C | 2k | 12,000 | 5,000 | 7,000 | - | 1,093 |
| | | 8k | 18,000 | 7,500 | 10,500 | - | 4,179 |
| | RepoBench-P | | 10,599 | 4,459 | 6,196 | 26 | 139,406 |

**RepoBench-R** RepoBench-R targets the retrieval component of a repository-level auto-completion system, focusing on extracting the most relevant code snippet from a project repository for next-line code prediction.

In RepoBench-R, every snippet parsed from import statements is treated as a potential candidate for next-line prediction, where only one 'gold snippet' is the optimal context for prediction. This task considers scenarios with 5 or more candidate snippets, and specifically, we categorize them into two subsets: those with 5-9 candidates as the *easy* subset, and those with 10 or more candidates as the *hard* subset. As demonstrated in Table 1, both the *easy* and *hard* subsets contain 12,000 samples for the XF-F setting, whereas for the XF-R setting, each subset consists of 6,000 samples. We also provide training data for optional usage, further details can be also located in Appendix B. For evaluative purposes, the Accuracy@k (acc@k) metric is employed to assess retrieval performance. The *easy* subset is evaluated using acc@1 and acc@3, while the *hard* subset is examined through acc@1, acc@3, and acc@5 metrics.

**RepoBench-C** RepoBench-C simply focuses on the prediction of the next line of code, given a set of in-file context (including several preceding lines and import statements), and cross-file context.

In RepoBench-C, as shown in Figure 1 the prompt is created by combining all the parsed snippets as cross-file contexts and an in-file context. The in-file context includes import statements and several preceding lines of code with a maximum limit of 30 lines. To address the varying context length in existing models, RepoBench-C is divided into two subsets: RepoBench-C-2k and RepoBench-C-8k. RepoBench-C-2k, designed for models with a token limit of 2,048, holds prompts that do not exceed 1,925 tokens. Concurrently, RepoBench-C-8k is architected with a higher threshold, encompassing up to 7,685 tokens, apt for models with an 8,192 token limit (e.g., StarCoder (Li et al., 2023)) or 8,000 token limit (e.g., Codex (Chen et al., 2021)).

RepoBench-C is designed primarily for 0-shot learning, in order to examine the model's ability to handle long-range contexts. Despite this, we also provide a large amount of training data to allow fine-tuning, thereby enhancing the transfer capabilities of relatively smaller models, and for the test set, we allocate more data under XF-F settings compared with XF-R and IF settings. Details of this data are provided in Table 1. For evaluation metrics, we adopt Exact Match (EM) and Edit Similarity (Edit Sim) (Svyatkovskiy et al., 2020b) following the previous work (Lu et al., 2021) and extend our evaluation with CodeBLEU (Ren et al., 2020) to evaluate the accuracy of the predicted code line.

**RepoBench-P** RepoBench-P evaluates the model's performance by combining RepoBench-R and RepoBench-C: retrieval of relevant snippets and next-line code prediction, presenting a challenging pipeline task. This task mirrors complex real-world scenarios that a practical auto-completion system would face, assessing the model's comprehensive performance and flexibility.

In RepoBench-P, each setting (XF-F, XF-R, and IF) requires the model to first identify the most pertinent snippets and then employ these snippets as cross-file context in conjunction with the in-file context to predict the subsequent line. Contrary to specifying a maximum token limit, we define

a minimum token threshold: 12,000 for Python and 24,000 for Java, and the gold snippet retrieval process requires a minimum of 10 candidates. Due to the substantial amount of data resulting from these constraints, we opt to down-sample to ensure parity between Java and Python datasets. Details of this data are provided in Table 1. For evaluating the predicted next line, we also use the Exact Match, Edit Similarity and CodeBLEU metrics, in line with the RepoBench-C setting.

## 4 EXPERIMENTS

### 4.1 REPOBENCH-R

The primary objective of the retrieval task in RepoBench-R is to identify the most relevant code snippets to predict the next line given an in-file context. The process generally involves cropping certain lines of the in-file code before the predicted line, followed by the calculation of the degree of relevance (we use the term 'similarity' uniformly) between the cropped code and each candidate snippet. Formally, the general method for retrieval can be mathematically formulated as follows:

$$\underset{i \in \{1,...,n\}}{\arg \max^{k}} f(C[-m :], S_i)$$

where $C$ denotes the in-file code, $S_i$ refers to the $i$-th candidate snippet, $n$ is the total number of candidate snippets, $m$ is the number of lines from the in-file code retained, $k$ represents the top $k$ candidates to be retrieved, and $f$ is the function computing the similarity (or other scores) between the cropped in-file code and the candidate snippets.

**Baseline** In our baseline approach, three strategies are employed for the retrieval task: (1) **Random Retrieval** involves retrieving code snippets in a random manner, serving as a lower-bound benchmark against which we can compare the effectiveness of the other retrieval methods. To ensure the stability and reliability of our results, this random process is repeated 100 times and the outcomes are averaged. (2) **Lexical Retrieval** uses Jaccard Similarity and Edit Similarity to assess the relevance between the cropped code from the in-file context and the candidate code snippets. (3) **Semantic Retrieval** applies encoder models, including CodeBERT (Feng et al., 2020) based on BERT (Devlin et al., 2019), UnixCoder (Guo et al., 2022) based on UniLM (Dong et al., 2019) ( we use the `Encoder-only` mode), and InstructOR (Su et al., 2023) to obtain the code embeddings. We also include some other encoder-decoder or decoder models, include CodeGPT (Lu et al., 2021), CodeT5+ (Wang et al., 2023) and CodeGen (Nijkamp et al., 2022). Cosine Similarity is employed to measure the semantic similarity between the cropped code and the candidate snippets. In baseline, we crop $m = 3$ lines from the in-file code as specified in the general method ($C[-m :]$), indicating the last three lines before the line to predict (Refer to the Appendix D for the ablation study on the number of lines kept). All computations for determining the similarity score are executed at the token level.

**Results and Analysis** Table 2 presents a detailed comparison of different retrieval strategies in RepoBench-R. We have the following observations: (1) **InstructOR outperforms retrieval models, followed by UniXcoder:** InstructOR consistently outperforms other retrieval models across tasks, with UniXcoder achieving comparable results despite having only 1/10 of InstructOR's parameters. This performance of UniXcoder can be partially attributed to its unique approach that includes multi-modal data representation learning and the utilization of both multi-modal contrastive learning (MCL) (Gao et al., 2021) and cross-modal generation tasks (CMG). (2) **Jaccard similarity offers a competitive lexical retrieval alternative:** Within the lexical retrieval category, Jaccard similarity has shown to be competitive, offering a viable and light-weighted alternative to semantic methods. (3) **Python code retrieval tasks yield higher accuracy than Java:** Tasks involving Python code retrieval tend to yield higher accuracy than those with Java, which is partially hypothesized to be due to the common Python practice of defining function arguments in close proximity to their calls, thereby providing valuable contextual cues for retrieval. In contrast, Java's extensive use of class structures may introduce additional complexities into the retrieval process.

### 4.2 REPOBENCH-C

The code completion task, RepoBench-C, aims to predict the next line of code based on a given in-file context ($C_{in}$), consisting of import statements and preceding lines before the target line, as

Table 2: Baseline weighted average results of RepoBench-R on Python and Java retrieval tasks for *Easy* and *Hard* subset. The models used are `codebert-base` for CodeBERT, `unixcoder-base` for UniXcoder, `CodeGPT-small-py` and `CodeGPT-small-small` for CodeGPT in Python and Java respectively, and `codegen-350M-mono` and `codegen-350M-multi` for CodeGen in Python and Java respectively, `codet5p-220m` for CodeT5+ and `instructor-xl` for InstructOR.

| Language | Retrieval | Model | Params. | Easy | | Hard | | |
| --- | --- | --- | --- | --- | --- | --- | --- | --- |
| | | | | acc@1 | acc@3 | acc@1 | acc@3 | acc@5 |
| Python | Random | - | - | 15.66 | 46.96 | 6.43 | 19.31 | 32.12 |
| | Lexical | Jaccard | - | 21.97 | 53.75 | 10.47 | 25.93 | 40.01 |
| | | Edit | - | 18.69 | 50.98 | 7.83 | 21.77 | 36.49 |
| | Semantic | CodeGPT | 124M | 16.18 | 47.05 | 8.27 | 22.79 | 36.35 |
| | | UniXcoder | 125M | 27.09 | 60.42 | 18.48 | 39.69 | **54.00** |
| | | CodeBERT | 125M | 16.94 | 48.27 | 6.72 | 19.89 | 33.05 |
| | | CodeT5+ | 220M | 18.32 | 50.95 | 8.58 | 23.03 | 36.24 |
| | | CodeGen | 350M | 21.03 | 54.27 | 13.20 | 31.64 | 46.22 |
| | | InstructOR | 1.5B | **28.22** | **62.77** | **19.10** | **39.91** | 53.54 |
| Java | Random | - | - | 15.36 | 46.03 | 5.61 | 16.89 | 28.16 |
| | Lexical | Jaccard | - | 16.58 | 48.49 | 7.84 | 20.83 | 32.80 |
| | | Edit | - | 15.19 | 45.92 | 6.09 | 17.63 | 28.69 |
| | Semantic | CodeGPT | 124M | 16.46 | 48.46 | 7.87 | 22.97 | 37.53 |
| | | CodeBERT | 125M | 15.68 | 46.51 | 6.02 | 17.52 | 28.80 |
| | | UniXcoder | 125M | 19.61 | 52.96 | 12.23 | 28.74 | 41.88 |
| | | CodeT5+ | 220M | 16.12 | 47.67 | 6.46 | 18.50 | 30.50 |
| | | CodeGen | 350M | **20.09** | 52.60 | 11.09 | 29.32 | 44.22 |
| | | InstructOR | 1.5B | 19.94 | **53.91** | **13.07** | **31.28** | **44.52** |

well as a cross-file context ($C_x$), comprising snippets from other files parsed by import statements. This task commonly uses autoregressive language models trained on code for prediction. The formal expression of this task can be illustrated as follows:

$$P(Y) = \prod_{i=1}^{n} P(y_i | y_{<i}, C_x, C_{in}) \tag{1}$$

where $P(Y)$ is the joint probability of all tokens in the predicted sequence $Y$. The variable $y_i$ denotes the $i^{th}$ token in sequence Y, while $y_{<i}$ symbolizes the sequence of all preceding tokens. $C_x$ and $C_{in}$ represent the cross-file context and the in-file context, respectively. This product notation represents the autoregressive assumption that each token $y_i$ is conditionally dependent on all preceding tokens $y_{<i}$ and the given contexts $C_x$ and $C_{in}$.

**Baseline** To establish a performance baseline, our benchmark compares 4 series of models: (1) **Codex**[4] (Chen et al., 2021) (i.e. `code-davinci-002`), developed by OpenAI, recognized for its code generation capabilities and serving as the base model for Copilot.(2) **CodeGen** (Nijkamp et al., 2022), a family of autoregressive language models for program synthesis, available in multiple size variants (350M, 2B, 6B, 16B). We use `CodeGen-Mono` for Python and `CodeGen-Multi` for Java in our evaluations. (3) **StarCoder** (Li et al., 2023), a 15.5B parameter model trained across over 80 programming languages. We adopt `StarCoder` for Python and `StarCoderBase` for Java. (4) **CodeLlama** (Rozière et al., 2024), a family of large language models for code based on Llama 2 (Touvron et al., 2023), include with 7B, 13B and 34B parameters. Similarly, we use `CodeLlama-Python` for Python and `CodeLlama` for Java.

**Results and Analysis** Table 7 presents the updated results of RepoBench-C. Our findings on the two RepoBench-C subsets provide several insights: (1) **CodeLlama-34B excels in Python for both 2k and 8k:** CodeLlama-34B achieves the highest performance across all metrics for Python code generation tasks in both the 2k and 8k subsets, outperforming Codex and other models. (2) **Codex**

---

[4]Codex has been decommissioned as of January 4th, 2024 and is no longer accessible.

Table 3: Performance comparison of models on RepoBench-C across Python and Java, using weighted average Exact Match (EM), Edit Similarity (Edit Sim), and CodeBLEU scores from three settings, for 2k (top) and 8k (bottom) subset.

| Model | Params. | Python | | | Java | | |
|---|---|---|---|---|---|---|---|
| | | EM | Edit Sim | CodeBLEU | EM | Edit Sim | CodeBLEU |
| CodeGen | 350M | 20.71 | 64.21 | 32.60 | 21.21 | 63.62 | 37.18 |
| CodeGen | 2.7B | 27.35 | 68.28 | 38.34 | 28.31 | 69.15 | 43.84 |
| CodeGen | 6.1B | 31.67 | 70.67 | 42.15 | 29.59 | 70.27 | 44.87 |
| CodeLlama | 7B | 34.10 | 71.24 | 43.46 | 35.80 | 76.75 | 49.98 |
| CodeLlama | 13B | 36.18 | 72.25 | 45.60 | 36.26 | 75.72 | 49.44 |
| StarCoder | 15.5B | 31.67 | 71.27 | 41.46 | 37.35 | 77.00 | 51.81 |
| CodeGen | 16.1B | 33.41 | 71.20 | 43.58 | 30.45 | 70.29 | 45.91 |
| CodeLlama | 34B | **37.40** | **72.98** | **47.04** | 39.41 | 78.52 | 53.56 |
| Codex | - | 31.31 | 72.21 | 41.45 | **42.47** | **80.01** | **55.62** |

| Model | Params. | Python | | | Java | | |
|---|---|---|---|---|---|---|---|
| | | EM | Edit Sim | CodeBLEU | EM | Edit Sim | CodeBLEU |
| CodeLlama | 7B | 33.24 | 70.44 | 43.14 | 33.45 | 74.33 | 47.64 |
| CodeLlama | 13B | 35.56 | 71.57 | 45.10 | 36.26 | 75.72 | 49.44 |
| StarCoder | 15.5B | 29.93 | 68.84 | 40.39 | 32.49 | 74.29 | 46.92 |
| CodeLlama | 34B | **36.26** | **72.19** | **45.71** | 36.84 | 76.06 | 50.77 |
| Codex | - | 32.13 | 71.89 | 42.27 | **40.52** | **77.97** | **53.63** |

**maintains dominance in Java for both 2k and 8k:** Despite the strong performance of CodeLlama and other models, Codex consistently emerges as the top-performing model for Java code generation tasks across both the 2k and 8k subsets. This demonstrates Codex's specialized capabilities for multilingual code generation.

## 4.3 REPOBENCH-P

RepoBench-P combines the retrieval and code completion tasks to form a pipeline, where the goal is to first retrieve the most relevant code snippets given an in-file context and then predict the optimal next line of code based on the retrieved snippets and the in-file context. This pipeline approach aims to leverage the strengths of both tasks to enhance code assistance systems' capabilities. The formal expression of RepoBench-P can be represented as follows:

$$P(Y) = \prod_{i=1}^{n} P(y_i | y_{<i}, S_1, \ldots, S_k, C_{in}) \tag{2}$$

where $P(Y)$ denotes the joint probability of all tokens in the predicted sequence $Y$. $y_i$ represents the $i$-th token in sequence $Y$, while $y_{<i}$ symbolizes the sequence of all preceding tokens. $S_1, \ldots, S_k$ refer to the retrieved code snippets, and $C_{in}$ represents the in-file context. This product notation signifies the autoregressive assumption that each token $y_i$ is conditionally dependent on all preceding tokens $y< i$, the given in-file context $C_{in}$, and the retrieved snippets $S_1, \ldots, S_k$.

**Baseline** To establish a performance baseline for the end-to-end task RepoBench-P, we test Codex (code-davinci-002) as the base model. We reserve 1,600 tokens for the in-file context, with a cropping limit of 60 preceding lines. Any unused tokens from this allocation are then filled by the cross-file context, up to a total prompt size of 6,400 tokens.

For the retrieval component, we delve into several strategies for retrieving relevant snippets from the cross-file context: (1) **Gold-Only**: In cross-file completions, the cross-file context includes just the 'gold snippet'. For in-file completions, the context is left empty. (2) **Gold-Filled**: The cross-file context integrates the 'gold snippet' alongside with other randomly fetched snippets until the 6,400-token capacity is filled. Inspired by the work of (Liu et al., 2023), we employ two variant strategies

Table 4: Comparison of various retrieval strategies on the RepoBench-P for Python and Java using Codex (Chen et al., 2021) (code-davinci-002). Each strategy is evaluated in terms of Exact Match (EM) and Edit Similarity (ES) metrics for XF-F, XF-R, and IF settings. 'All' represents the average performance over the mixture of all test data, weighted by the size of each test setting. Strategies (*Gold-Only* and *Gold-Filled*), marked with an asterisk (*), include gold snippets for benchmarking purposes and serve only as references; they do not embody oracle capabilities.

| Retrieval Method | Python | | | Java | | |
|---|---|---|---|---|---|---|
| | EM | Edit Sim | CodeBLEU | EM | Edit Sim | CodeBLEU |
| Baseline | 33.15 | 72.19 | 44.07 | 40.40 | 74.28 | 56.64 |
| Random | 34.94 | 73.10 | 45.89 | 40.77 | 74.51 | 56.88 |
| Jaccard | 36.46 | 73.66 | 46.76 | 41.48 | **74.93** | **57.39** |
| UniXcoder-H2L | 36.61 | 74.02 | **47.28** | 41.70 | 74.82 | 57.29 |
| UniXcoder-L2H | **37.11** | **74.31** | 47.15 | **41.83** | 74.93 | 57.12 |
| Gold-Only* | 35.79 | 73.58 | 46.55 | 41.62 | 74.95 | 57.09 |
| Gold-Filled-Head* | 36.07 | 73.67 | 46.90 | 41.59 | 74.88 | 56.99 |
| Gold-Filled-Tail* | 36.18 | 73.76 | 46.75 | 41.49 | 74.91 | 57.09 |

for the placement of the 'gold snippet': *Gold-Filled-Head*, where the 'gold snippet' is positioned at the beginning; and *Gold-Filled-Tail*, where it is positioned at the tail-end. (3) **UniXcoder**: Using UniXcoder as cross-file context retriever, snippets are obtained based on their relevance to the cropped preceding three in-file lines while adhering to a 6,400-token limit for the input length. The includes the *UniXcoder-H2L (High-to-Low)* variant, ranking snippets from the most to least relevant, and the *UniXcoder-L2H (Low-to-High)* approach, ranking in the reverse order. (5) **Random**: Cross-file context snippets are totally randomly selected without considering their relevance until the token limit is reached. Due to the constraints imposed by the codex rate limit, we are unable to perform multiple runs for the random retrieval, which is necessary to mitigate the inherent randomness; consequently, the results presented should be considered indicative and not conclusive. (6) **Baseline**: In this strategy, a token limit of 6,400 is solely allocated for the in-file context, abstaining from using any cross-file context during completion. It serves as a fundamental point of comparison, highlighting the model's performance when exclusively dependent on the in-file context.

**Results and Analysis** Table 4 presents a comparison of various retrieval strategies using Codex in RepoBench-P. From this comparison, we present the following insights: (1) **Inclusion of cross-file contexts improves performance:** Integrating more cross-file contexts enhances performance, irrespective of retrieval quality. Even randomly selected contexts significantly boost performance, potentially by fostering contextual understanding, enabling the model to draw from a broader code repository. (2) **Effective retrieval enhances performance:** Retrievers deploying specific models or methods like UniXcoder model, outperform random retrieval systems. Notably, this improvement is not confined to cross-file line prediction (XF-F and XF-R) but is also observed in in-file next-line prediction (IF), highlighting the value of retrieving code related to current code in the same repository as cross-file contexts, even if the succeeding line does not encompass cross-file modules. (3) **Placement order of retrieved code snippets matters:** The positioning of related code snippets influences code completion effectiveness. Positioning higher similar code snippets adjacent to or in close proximity to the line requiring completion tends to improve code completion performance.

## 5 CONCLUSION

In this paper, RepoBench is introduced as a benchmark designed for evaluating repository-level code auto-completion systems, comprising three distinct yet interrelated tasks: RepoBench-R for code retrieval, RepoBench-C for code completion, and RepoBench-P for testing the complete auto-completion pipeline. These tasks collectively provide a diverse evaluation environment for the Python and Java programming languages. The paper underscores through evaluation experiments the need for models capable of handling longer and more complex contexts, akin to those encountered in real-world programming scenarios. RepoBench aims to serve as a live benchmark that contributes to ongoing innovation in code intelligence.

## ACKNOWLEDGMENTS

We sincerely thank Zhiting Hu for his invaluable support and Daya Guo for her insightful suggestions and thorough proofreading of our manuscript.

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

# A    ABLATION STUDY FOR PROMPT CONSTRUCTION

In this appendix, we present a pilot study that focuses on constructing appropriate prompts for cross-file code completion. Our study is conducted using Codex, specifically, `code-davinci-002`, in an ablation setting where we systematically vary the prompt design. The following elements constitute the building blocks of our prompts:

1. **In-File Context (IFC)**: In-file contexts are the preceding $n$ lines before the line we want to predict. We investigate two cases - **Short IFC (IFC-Short)** which crops a maximum of the preceding 30 lines, and **Long IFC (IFC-Long)** which crops a maximum of the preceding 120 lines.

2. **Import Statements (IS)**: To avoid losing the import statements due to cropping, we construct the prompt by concatenating all the IS before the IFC.

3. **Cross-File Context (XFC)**: Cross-file contexts are commented code snippets from other files parsed from import statements. We attach them at the beginning of the prompt.

Table 5: Ablation study comparing different combinations of Cross-File Context (XFC), Import Statements (IS), and In-File Context (IFC) with both short (IFC-Short) and long (IFC-Long) variants, using Codex (Chen et al., 2021). The All score of Exact Match (EM) and Edit Similarity (ES) are calculated by averaging the three settings. The results are based on examples that are randomly sampled from the training set of RepoBench-C, with 5,000 examples for each of the three settings (XF-F, XF-R, IF) for Python and Java separately.

| | Prompt Construction | XF-F | | XF-R | | IF | | All | |
|---|---|---|---|---|---|---|---|---|---|
| | | EM | ES | EM | ES | EM | ES | EM | ES |
| Python | IFC-Short | 7.30 | 58.38 | 36.28 | 76.70 | 36.34 | 75.68 | 26.64 | 70.25 |
| | IFC-Long | 7.96 | 59.37 | 43.44 | 80.16 | 38.56 | 76.62 | 29.99 | 72.05 |
| | IS+IFC-Short | 22.28 | 69.30 | 38.90 | 78.03 | 38.46 | 77.25 | 33.21 | 74.86 |
| | IS+IFC-Long | 23.62 | 69.99 | **44.82** | **80.44** | 40.80 | 78.16 | 36.42 | 76.20 |
| | XFC+IFC-Short | 19.64 | 66.44 | 41.62 | 78.86 | 39.12 | 77.04 | 33.32 | 74.11 |
| | XFC+IS+IFC-Short | **28.38** | **72.46** | 43.30 | 79.68 | **41.24** | **78.29** | **37.64** | **77.06** |
| Java | IFC-Short | 7.00 | 57.41 | 33.30 | 74.57 | 57.76 | 80.94 | 32.69 | 70.97 |
| | IFC-Long | 7.32 | 57.95 | 37.12 | 76.74 | 59.74 | 81.76 | 34.73 | 72.15 |
| | IS+IFC-Short | 22.72 | 71.72 | 37.92 | 77.73 | 60.84 | 83.29 | 40.49 | 77.58 |
| | IS+IFC-Long | 23.36 | 72.34 | 40.68 | 79.01 | 62.98 | 84.48 | 42.34 | 78.61 |
| | XFC+IFC-Short | 23.58 | 68.02 | 41.86 | 79.25 | 60.42 | 82.29 | 41.95 | 76.52 |
| | XFC+IS+IFC-Short | **31.80** | **75.03** | **44.62** | **81.21** | **63.44** | **84.57** | **46.62** | **80.27** |

We consider 6 different combinations, as shown in Table 5. Our pilot study yields several notable results regarding cross-file code completion, as summarized below:

1. The integration of both ISC and XFC shows the best overall results and significantly enhances cross-file code completion performance, even though there may be duplicated information between XFC and ISC.

2. Including ISC and XFC improves not only cross-file completion but also in-file completion. This improved performance is observed even when the included snippets do not specifically target in-file completion.

3. For XF-R settings, where the module that the next line will use is possibly used in the IFX, which may provide hints for next-line prediction, the inclusion of a longer in-file context (IF-Long) appears to be beneficial for Python. This suggests that extended context within the same file can potentially help prediction if it is not the first usage. However, the combination of IFC with IS and XFC (XFC+IS+IFC) yields nearly comparable results, which highlights the role of cross-file context and import statements.

Thus, as shown in Figure 1, in our dataset, we leverage the prompt construction by adopting the XFC+IS+IFC-S strategy, incorporating both cross-file context, import statements and short in-file context as the input for LLMs.

Table 6: Training data for RepoBench across Python and Java.

| Language | Task | XF-F | XF-R | IF |
|---|---|---|---|---|
| Python | Code Retrieval | 175,199 | 86,180 | - |
| | Code Completion | 349,023 | 179,137 | 214,825 |
| java | Code Retrieval | 340,121 | 216,642 | - |
| | Code Completion | 683,890 | 447,464 | 709,218 |

## B  TRAINING DATA

Table 6 summarizes the optional training data numbers for RepoBench.

## C  REPOBENCH V1.1

### C.1  INTRODUCTION

In this appendix, we provide a description of the newest version of RepoBench v1.1[5,6] as of the writing. This version of RepoBench, also including datasets for Python and Java, was constructed from GitHub data spanning from `October 6, 2023,` to `December 31, 2023.` To further address potential concerns of data leakage and memorization, which could bias model evaluations, we also performed a deduplication process on the Stack v2 (Lozhkov et al., 2024).

In RepoBench v1.1, to accommodate models capable of handling longer inputs and to streamline the evaluation process, we have reduced the volume of the test dataset and categorized the next-line prediction tasks into distinct levels based on their prompt lengths utilizing OpenAI's GPT-4 tokenizer for tokenization. The levels are determined by specific token ranges, including `Level 2k` ($640 \sim 1,600$ tokens), `Level 4k` ($1,600 \sim 3,600$ tokens), `Level 8k` ($3,600 \sim 7,200$ tokens), `Level 12k` ($7,200 \sim 10,800$ tokens), `Level 16k` ($10,800 \sim 14,400$ tokens), `Level 24k` ($14,400 \sim 21,600$ tokens), `Level 32k` ($21,600 \sim 28,800$ tokens), `Level 64k` ($28,800 \sim 57,600$ tokens), and `Level 128k` ($57,600 \sim 100,000$ tokens).

### C.2  EVALUATION

RepoBench v1.1 is featured in the StarCoder 2 (Lozhkov et al., 2024) technical report as a benchmark for evaluating the capabilities of base models for repository-level code completion. It includes evaluations at 5 levels, specifically 2k, 4k, 8k, 12k, and 16k, to accommodate the 16k sequence length for most of the current open-source models. Table 7 presents the performance of open-access base models on RepoBench v1.1.

In comparison, it is noteworthy that StarCoder2 exhibits a significant improvement over StarCoder1, which can be attributed to two main factors. Firstly, the increase in context length has likely contributed to better model performance. More importantly, however, is the pre-training of StarCoder2 at the repository-level, underscoring the essential role of repository-level training in enhancing model capabilities.

## D  ABLATION STUDY OF KEPT LINES FOR RETRIEVAL

In this appendix, an ablation study is conducted to ascertain the optimal number of preceding lines to be retained during the retrieval of pertinent code snippets for the prediction line. The study evaluates four distinct retrieval methodologies, namely, two Lexical Retrievers (*Jaccard Similarity* and *Edit Similarity*) and two Semantic Retrievers (*CodeBERT* and *UniXcoder*). The experimental evaluation considers keeping 3, 5, 10, 20, 30, 60, and 120 lines for the retrieval process. For consistency, the model sizes selected for this ablation study align with the configurations delineated in Section 4.1.

---

[5]`https://huggingface.co/datasets/tianyang/repobench_python_v1.1`
[6]`https://huggingface.co/datasets/tianyang/repobench_java_v1.1`

Table 7: Average exact match (EM), edit similarity (Edit Sim), and CodeBLEU scores for open-access base models on RepoBench v1.1. (Table copy from StarCoder 2 technical report (Lozhkov et al., 2024))

| Model | Python | | | Java | | |
|---|---|---|---|---|---|---|
| | EM | Edit Sim | CodeBLEU | EM | Edit Sim | CodeBLEU |
| StarCoderBase-3B | 29.99 | 69.37 | 36.77 | 36.01 | 74.18 | 45.30 |
| DeepSeekCoder-1.3B | 31.02 | 70.07 | 37.88 | 37.75 | 75.66 | 46.69 |
| StableCoder-3B | **34.48** | **71.79** | **40.43** | **40.13** | **76.56** | **49.00** |
| StarCoder2-3B | 32.47 | 71.19 | 39.25 | 38.46 | 76.53 | 47.96 |
| StarCoderBase-7B | 32.70 | 71.08 | 39.48 | 37.97 | 75.66 | 47.47 |
| CodeLlama-7B | 33.85 | 71.79 | 40.47 | 39.61 | 76.71 | 48.92 |
| DeepSeekCoder-6.7B | **36.79** | **73.85** | **42.65** | **42.87** | **78.93** | **51.69** |
| StarCoder2-7B | 33.72 | 72.07 | 40.34 | 39.84 | 77.23 | 48.96 |
| StarCoderBase-15B | 33.51 | 71.64 | 40.39 | 39.34 | 76.24 | 48.36 |
| CodeLlama-13B | 35.50 | 72.98 | 42.02 | 41.27 | 77.57 | 50.26 |
| StarCoder2-15B | **36.99** | **74.08** | **43.25** | **42.57** | **79.05** | **51.45** |
| CodeLlama-34B | 37.22 | 73.77 | 43.38 | 42.35 | 78.22 | 50.99 |
| DeepSeekCoder-33B | **39.25** | **75.20** | **45.21** | **44.59** | **79.92** | **52.70** |

Table 8: Performance of Jaccard Similarity as retrieval method for different numbers of kept lines.

| | Lines | Easy Level | | | | Hard Level | | | | | |
|---|---|---|---|---|---|---|---|---|---|---|---|
| | | XF-F | | XF-R | | XF-F | | | XF-R | | |
| | | acc@1 | acc@3 | acc@1 | acc@3 | acc@1 | acc@3 | acc@5 | acc@1 | acc@3 | acc@5 |
| Python | *Rand* | 15.72 | 47.16 | 15.79 | 47.08 | 6.67 | 20.00 | 33.41 | 6.79 | 20.28 | 33.67 |
| | 3 | **18.10** | **51.38** | 24.40 | 55.73 | 11.24 | 30.04 | **45.23** | 15.43 | 35.25 | 49.53 |
| | 5 | 17.94 | 50.72 | **24.68** | 56.27 | **11.51** | **30.66** | 44.96 | 16.70 | 36.12 | 49.38 |
| | 10 | 17.65 | 49.83 | 24.60 | **56.60** | 11.29 | 29.07 | 44.44 | **17.35** | **37.03** | **50.60** |
| | 20 | 16.79 | 48.01 | 22.43 | 55.55 | 10.40 | 27.15 | 41.56 | 15.65 | 34.42 | 48.35 |
| | 30 | 16.18 | 47.93 | 22.07 | 54.10 | 9.09 | 25.10 | 39.64 | 14.25 | 32.25 | 46.23 |
| | 60 | 15.61 | 46.29 | 19.38 | 52.25 | 8.03 | 23.03 | 36.93 | 12.30 | 28.88 | 42.83 |
| | 120 | 15.02 | 45.67 | 18.32 | 50.88 | 7.03 | 21.43 | 35.04 | 10.45 | 26.27 | 40.30 |
| Java | *Rand* | 15.81 | 47.52 | 15.82 | 47.40 | 6.92 | 20.79 | 34.68 | 6.95 | 20.90 | 34.90 |
| | 3 | **16.98** | **49.89** | **21.32** | 52.90 | **9.14** | **25.97** | **41.39** | 12.15 | 29.70 | 43.45 |
| | 5 | 16.53 | 49.06 | 21.22 | **54.30** | 8.92 | 25.45 | 40.34 | **12.25** | **30.05** | 43.35 |
| | 10 | 15.82 | 46.54 | 21.25 | 54.20 | 8.06 | 24.20 | 38.42 | 12.07 | 29.12 | **43.50** |
| | 20 | 14.32 | 45.65 | 19.50 | 53.67 | 6.54 | 21.91 | 36.52 | 11.28 | 28.52 | 43.12 |
| | 30 | 14.05 | 45.02 | 18.88 | 52.38 | 6.31 | 20.95 | 35.80 | 11.12 | 28.35 | 42.95 |
| | 60 | 13.55 | 43.94 | 17.72 | 50.92 | 6.05 | 19.30 | 34.19 | 9.57 | 26.65 | 41.17 |
| | 120 | 13.21 | 43.14 | 17.20 | 49.45 | 5.51 | 18.38 | 33.10 | 9.00 | 25.50 | 39.73 |

The experiments are executed on subsets of the training data, encompassing 8,000 XF-F samples and 4,000 XF-R samples, each categorically divided into easy and hard subsets. The accompanying tables delineate the performance metrics for each retrieval method: *Jaccard Similarity* (Table 8), *Edit Similarity* (Table 9), *CodeBERT* (Table 10), and *UniXcoder* (Table 11). The ablation analysis aims to elucidate the influence of varying the number of retained lines on the performance of code retrieval in the code generation task. The empirical results suggest a general trend: as the number of retained lines increases, the performance of most retrievers tends to diminish, with the optimal performance typically achieved when retaining either 3 or 5 lines.

Table 9: Performance of Edit Similarity as retrieval method for different numbers of kept lines.

| | Lines | Easy Level | | | | Hard Level | | | | | |
| | | XF-F | | XF-R | | XF-F | | | XF-R | | |
| | | acc@1 | acc@3 | acc@1 | acc@3 | acc@1 | acc@3 | acc@5 | acc@1 | acc@3 | acc@5 |
|---|---|---|---|---|---|---|---|---|---|---|---|
| Python | *Rand* | 15.72 | 47.16 | 15.79 | 47.08 | 6.67 | 20.00 | 33.41 | 6.79 | 20.28 | 33.67 |
| | 3 | **16.56** | **48.74** | **20.38** | 51.05 | **9.54** | **25.60** | **39.77** | **11.97** | 28.10 | 41.83 |
| | 5 | 16.53 | 48.59 | 20.08 | 50.75 | 9.40 | 25.06 | 39.51 | 11.58 | **28.95** | **42.45** |
| | 10 | 16.20 | 47.09 | 19.78 | 51.20 | 8.54 | 23.79 | 38.20 | 10.47 | 27.27 | 42.50 |
| | 20 | 15.41 | 45.81 | 18.75 | **52.12** | 7.98 | 22.35 | 36.54 | 9.98 | 26.05 | 41.58 |
| | 30 | 15.64 | 46.11 | 18.85 | 51.05 | 6.95 | 21.45 | 35.93 | 9.40 | 24.85 | 39.88 |
| | 60 | 14.49 | 44.73 | 18.65 | 49.30 | 6.78 | 20.12 | 33.66 | 8.03 | 23.05 | 36.83 |
| | 120 | 14.03 | 44.64 | 16.98 | 48.12 | 6.35 | 19.78 | 33.52 | 8.15 | 21.88 | 35.20 |
| Java | *Rand* | 15.81 | 47.52 | 15.82 | 47.40 | 6.92 | 20.79 | 34.68 | 6.95 | 20.90 | 34.90 |
| | 3 | **16.65** | **49.14** | 16.07 | 48.43 | 7.21 | **22.44** | **36.88** | **8.20** | **22.73** | **36.62** |
| | 5 | 16.04 | 48.33 | 16.28 | 48.25 | 7.12 | 22.15 | 36.12 | **8.20** | 21.82 | 35.60 |
| | 10 | 15.64 | 46.90 | **16.40** | **49.05** | 6.64 | 20.77 | 34.98 | 7.95 | 21.50 | 35.38 |
| | 20 | 15.78 | 46.60 | 16.10 | 47.90 | 6.81 | 20.23 | 34.25 | 7.15 | 21.93 | 35.15 |
| | 30 | 15.15 | 46.30 | 15.68 | 47.48 | 6.73 | 20.21 | 34.11 | 7.45 | 21.43 | 34.67 |
| | 60 | 15.01 | 45.91 | 14.80 | 46.30 | 6.10 | 19.38 | 33.17 | 6.48 | 19.68 | 33.85 |
| | 120 | 14.97 | 45.96 | 14.62 | 46.27 | 6.19 | 19.19 | 32.17 | 6.48 | 20.23 | 33.62 |

Table 10: Performance of CodeBERT (Feng et al., 2020) (`codebert-base`) as the retriever for different numbers of kept lines.

| | Lines | Easy Level | | | | Hard Level | | | | | |
| | | XF-F | | XF-R | | XF-F | | | XF-R | | |
| | | acc@1 | acc@3 | acc@1 | acc@3 | acc@1 | acc@3 | acc@5 | acc@1 | acc@3 | acc@5 |
|---|---|---|---|---|---|---|---|---|---|---|---|
| Python | *Rand* | 15.72 | 47.16 | 15.79 | 47.08 | 6.67 | 20.00 | 33.41 | 6.79 | 20.28 | 33.67 |
| | 3 | **15.81** | 46.36 | **15.90** | 45.95 | 6.83 | 20.66 | **34.52** | **7.20** | 19.85 | 33.60 |
| | 5 | 15.36 | 46.48 | 15.40 | 46.02 | **7.19** | **21.01** | 34.39 | 6.42 | 19.82 | 33.23 |
| | 10 | 15.41 | 46.17 | 15.55 | 45.85 | 6.86 | 20.21 | 34.17 | 5.92 | 19.82 | 32.75 |
| | 20 | 15.39 | 46.35 | 15.62 | 46.48 | 6.85 | 19.86 | 33.39 | 7.05 | **21.62** | 33.90 |
| | 30 | 14.91 | 46.65 | 15.62 | **47.10** | 6.84 | 20.11 | 33.19 | 6.90 | 20.47 | **33.95** |
| | 60 | 14.72 | 46.73 | 14.95 | 47.05 | 6.53 | 19.51 | 32.96 | 6.30 | 19.50 | 32.60 |
| | 120 | 14.71 | **46.74** | 15.02 | 46.98 | 6.62 | 19.35 | 32.80 | 6.60 | 20.42 | 33.05 |
| Java | *Rand* | 15.81 | 47.52 | 15.82 | 47.40 | 6.92 | 20.79 | 34.68 | 6.95 | 20.90 | 34.90 |
| | 3 | 16.21 | 48.35 | 15.90 | 47.67 | 7.38 | **22.26** | 36.99 | 6.55 | 20.20 | 34.40 |
| | 5 | **16.26** | **48.73** | 16.68 | 47.93 | **7.58** | 21.79 | 36.16 | 6.50 | 20.42 | 34.88 |
| | 10 | 15.35 | 48.19 | 16.48 | 48.25 | 7.00 | 21.50 | 35.62 | **7.20** | 20.62 | 34.40 |
| | 20 | 15.79 | 47.95 | **17.35** | 48.30 | 7.15 | 20.94 | 34.39 | 7.15 | 21.00 | **36.38** |
| | 30 | 15.78 | 47.58 | 17.25 | **48.40** | 6.61 | 20.86 | 34.81 | 6.73 | 21.00 | 35.68 |
| | 60 | 15.05 | 47.70 | 16.43 | 48.10 | 6.21 | 20.86 | 34.51 | 6.88 | **21.77** | 35.95 |
| | 120 | 14.80 | 47.21 | 16.28 | 48.33 | 6.15 | 20.31 | 34.04 | 6.73 | 21.02 | 35.48 |

Table 11: Performance of UniXcoder (Guo et al., 2022) (`unixcoder-base`) as the retriever for different numbers of kept lines.

| | Lines | Easy Level | | | | Hard Level | | | | | |
| | | XF-F | | XF-R | | XF-F | | | XF-R | | |
| | | acc@1 | acc@3 | acc@1 | acc@3 | acc@1 | acc@3 | acc@5 | acc@1 | acc@3 | acc@5 |
|---|---|---|---|---|---|---|---|---|---|---|---|
| Python | *Rand* | 15.72 | 47.16 | 15.79 | 47.08 | 6.67 | 20.00 | 33.41 | 6.79 | 20.28 | 33.67 |
| | 3 | **27.02** | **60.14** | 29.23 | 61.95 | **19.86** | **41.40** | 55.74 | 22.50 | 43.12 | 56.93 |
| | 5 | 25.99 | 59.96 | **29.62** | **62.78** | 19.38 | 41.30 | **55.75** | **22.55** | **44.98** | **58.83** |
| | 10 | 23.34 | 57.53 | 27.80 | 61.02 | 16.89 | 39.29 | 53.73 | 20.93 | 42.83 | 56.53 |
| | 20 | 20.20 | 54.47 | 24.25 | 56.62 | 12.83 | 32.10 | 47.00 | 16.38 | 36.27 | 50.62 |
| | 30 | 18.68 | 51.95 | 21.48 | 54.05 | 10.82 | 27.56 | 42.12 | 12.95 | 31.32 | 46.25 |
| | 60 | 17.05 | 48.65 | 19.07 | 51.02 | 8.03 | 22.82 | 36.61 | 10.30 | 26.75 | 40.27 |
| | 120 | 16.09 | 47.38 | 18.12 | 48.83 | 7.12 | 20.51 | 33.73 | 8.33 | 22.65 | 34.67 |
| Java | *Rand* | 15.81 | 47.52 | 15.82 | 47.40 | 6.92 | 20.79 | 34.68 | 6.95 | 20.90 | 34.90 |
| | 3 | 20.44 | **54.25** | **27.25** | 61.48 | **13.79** | **35.73** | **51.54** | **21.05** | 43.38 | 57.80 |
| | 5 | 19.24 | 53.01 | 26.47 | **61.65** | 13.18 | 33.98 | 50.18 | 20.15 | **43.40** | **58.17** |
| | 10 | 16.16 | 49.58 | 25.55 | 60.12 | 11.06 | 30.18 | 46.16 | 17.70 | 40.65 | 56.62 |
| | 20 | 14.69 | 46.64 | 21.93 | 57.98 | 8.96 | 26.20 | 41.90 | 14.80 | 35.75 | 51.45 |
| | 30 | 13.81 | 45.40 | 20.25 | 56.15 | 7.96 | 24.27 | 39.40 | 13.40 | 33.12 | 48.62 |
| | 60 | 12.85 | 43.66 | 17.47 | 51.25 | 6.40 | 20.72 | 34.79 | 9.35 | 26.57 | 42.33 |
| | 120 | 12.31 | 42.50 | 16.50 | 48.62 | 5.76 | 18.79 | 32.67 | 7.90 | 22.53 | 37.50 |

# E    EXPERIMENT SETTINGS

All models (except codex) use CTranslate2 (OpenNMT, 2023) for inference [7] and the model weights are sourced from Huggingface (Wolf et al., 2020).The Codex model was accessed via OpenAI's API with authorized usage rights before its deprecation as of March 2023. The results for Codex were obtained through queries conducted from January 2023 to September 2023. During inference for new token generation, all models are set with a temperature of 0.2 and a top_p of 0.95, generating 64 tokens per next-line prediction, with the first non-comment line truncated as the output.

---

[7]Due to limited experimental resources and the extensive scale of our experiments, we rely on quantized models and libraries known for fast inference speeds. This reliance potentially introduces discrepancies in our results from the orginal models due to quantization effects or bugs within these libraries.

