# OpenReview forum: "RepoBench: Benchmarking Repository-Level Code Auto-Completion Systems"
_ICLR.cc/2024/Conference — ICLR 2024 poster_

### Official Review · Reviewer_mVr1 · 2023-10-30

**Soundness:** 3 good
**Presentation:** 3 good
**Contribution:** 3 good
**Rating:** 5
**Confidence:** 3

**Summary:**

This paper proposes a new benchmark called RepoBench for evaluating repository-level code completion systems. RepoBench consists of three interconnected evaluation tasks: RepoBench-R for retrieving the most relevant code in the repository, RepoBench-C for code completion using both in-file and cross-file context, and RepoBench-P for the entire pipeline of both retrieval and code completion. The authors carry out a series of experiments on RepoBench, analyzing the efficacy of various retrieval methods and code completion models of different magnitudes.

**Strengths:**

- A nice idea of benchmarking repository-level code completion
 - The paper is generally well-written and easy to follow

**Weaknesses:**

To me, the novel research contributions of the paper are a bit limited, especially for an AI conference. The paper could better fit a software engineering/programming conference.  The significance of the work could be more clearly stated.

To evaluate RepoBench-R, the authors selected three baseline strategies for the retrieval task, namely, random retrieval, lexical retrieval, and semantic retrieval. The selection of baseline strategies for RepoBench-R, particularly the inclusion of random retrieval and lexical retrieval, are weak baselines, which may not effectively demonstrate the distinctive capabilities of the proposed benchmark. In that sense, the results presented in Section 4.1 are under expectation and I think that previous benchmarks may also demonstrate the ability of these strategies. A more competitive baseline selection including LLMs would enhance the work.

The paper lacks a comprehensive comparison with previous benchmarks about code completion. Although RepoBench is the first benchmark on repository-level code completion, it would still benefit from comparisons with prior benchmarks. Such comparisons could involve RepoBench-R versus existing code retrieval benchmarks and RepoBench-C versus traditional benchmarks for function-level code completion.

The metrics used for code completion, i.e., EM and Edit Similarity, are unusual. The authors could consider more widely used metrics such as pass@k and CodeBLEU?

The evaluation of RepoBench-C is conducted using only three Language Model Models (LLMs), specifically CodeGen, StarCoder, and Codex. As a benchmark paper, the inclusion of only three LLMs may not fully represent the diverse capabilities of available models. To enhance the benchmark's applicability, additional LLMs, including recently proposed ones, could be considered for comparison. For example: Shi et al., SoTaNa: The Open-Source Software Development Assistant, https://arxiv.org/abs/2308.13416

**Questions:**

- Why not using widely used metrics such as pass@k and CodeBLEU?
- How is the proposed benchmark compared to previous benchmarks for code completion?

---

> ### Author Response · Authors · 2023-11-22
> **Response to Reviewer mVr1 (1/2)**
>
> Dear Reviewer mVr1,
>
> We appreciate the time you invested in reviewing our paper. Your comments have provided valuable perspectives that we have carefully considered in this response.
>
> > The novel research contributions of the paper are a bit limited, especially for an AI conference.
>
> We would like to highlight the significance of our work:
>
> 1. We hope that RepoBench can address the evaluation of code generation in LLMs at current stage. This is a key area in the field of AI, as understanding and improving code generation capabilities are vital.
> 2. RepoBench-C can be used to facilitate tasks involving long-context generation. The customizable token number feature of RepoBench-C allows the construction of 2k, 8k, 16k and even 128k versions, making it a benchmark for evaluating the efficacy of LLMs in handling long context length. This flexibility is particularly important in the current landscape of AI research, where handling longer contexts is becoming increasingly crucial.
> 3. We are encouraged by the interest RepoBench has already garnered in the LLM community. The StarCoder team has noticed the performance of StarCoder here, leading to a collaboration study focused on repo-level model training for next version. You can check the preliminary ablation study we already have shown in the global response.
>
> >  Weak baselines selection of RepoBench-R.
>
> First, we would like to emphasize the relevance of Jaccard similarity as a baseline. It has been revealed through reverse engineering studies online that GitHub Copilot uses Jaccard similarity backend for retrieval.
>
> In our initial experiments, we evaluated both encoder-only and decoder models, such as CodeGPT and CodeGen. While encoder-only models demonstrated superior performance for retrieval tasks, we initially omitted these results due to page constraints. In response to your feedback, we plan to include these findings to present a more comprehensive evaluation.
>
> We are also excited to share that we have conducted additional experiments with more recent encoding models, specifically CodeT5+ and Intruct-XL. The results from these experiments have shown **Instruct-XL demonstrates similar performance to UnixCoder**.
>
> Table 4: Results of CodeGPT, CodeGen, CodeT5+ and Instruct-XL as retriever on RepoBench-R.
>
> |       Language |  Model     | acc@1 (Easy XF-F) | acc@3 (Easy XF-F) | acc@1 (Easy XF-R) | acc@3 (Easy XF-R) | acc@1 (Hard XF-F) | acc@3 (Hard XF-F) | acc@5 (Hard XF-F) | acc@1 (Hard XF-R) | acc@3 (Hard XF-R) | acc@5 (Hard XF-R) |
> |------------------|-----------|-------------------|-------------------|-------------------|-------------------|-------------------|-------------------|-------------------|-------------------|-------------------|-------------------|
> | **Python**       |     CodeGPT       | 15.86 | 47.12 | 16.83 | 46.92 | 8.46 | 23.15 | 36.71| 7.90 | 22.07 | 35.62 |
> | | CodeGen | 20.15 | 53.90 | 22.78 | 55.02 | 12.81 | 30.89 | 45.96| 13.98 | 33.15 | 46.73 |
> | | CodeT5+ | 17.86 | 50.61 | 19.25 | 51.63 | 8.39 | 22.53 | 36.02| 8.97 | 24.02 | 36.68 |
> | | Instruct-XL | 26.37 | 61.64 | 31.92 | 65.03 | 18.12 | 39.12 | 52.62| 21.07 | 41.50 | 55.38 |
> | **Java**       |     CodeGPT       | 15.89 | 48.09 | 17.60 | 49.20 | 7.81 | 23.31 | 37.59| 7.98 | 22.30 | 37.40 |
> | | CodeGen | 18.49 | 51.84 | 23.28 | 54.12 | 9.99 | 28.30 | 43.06| 13.30 | 31.35 | 46.55 |
> | | CodeT5+ | 15.73 | 46.75 | 16.90 | 49.52 | 6.22 | 17.93 | 29.66| 6.95 | 19.65 | 32.17 |
> | | Instruct-XL | 16.88 | 50.58 | 26.05 | 60.57 | 10.69 | 28.09 | 41.58| 17.83 | 37.67 | 50.40 |

---

> > ### Author Response · Authors · 2023-11-22
> > **Response to Reviewer mVr1 (2/2)**
> >
> > > The paper lacks a comprehensive comparison with previous benchmarks about code completion.
> >
> > Thank you for highlighting the need for a more thorough comparison with previous benchmarks. While our paper does include some discussion on 'code completion datasets' in related work section (Section 2), we acknowledge that this discussion might be somewhat limited, and we appreciate your guidance on enhancing this aspect of our work.
> >
> > Firstly, as you pointed out, RepoBench distinguishes itself by focusing on repository-level tasks, which we believe is not only practical but also closely aligns with real-world applications. Also, we focus on using the most recent repositories to mitigate the issue of data leakage. In contrast to benchmarks that might focus on narrower datasets, RepoBench encompasses thousands of repositories. This extensive coverage is critical in evaluating modern models, especially considering that some may employ strategies like 'rephrasing' to achieve high scores on benchmarks like HumanEval. Our broad dataset scope ensures a more thorough and diverse evaluation.
> >
> > Unlike previous retrieval tasks that often focus on simple semantic matching, our benchmark's code retrieval is contextualized within the code completion process. This adds a layer of practical significance, especially for projects aimed at enhancing code completion tools.
> >
> > In light of your valuable suggestion, we plan to include a more detailed comparison of both retrieval and completion aspects in our revised paper.
> >
> > > Why not using widely used metrics such as pass@k and CodeBLEU?
> >
> > Firstly, our selection of EM and ES metrics simply aligns with  prior research on code completion. For instance, [CodeXGLUE](https://arxiv.org/pdf/2102.04664.pdf), [LongCoder](https://arxiv.org/abs/2306.14893).
> >
> > Regarding pass@k, we acknowledge its potential as an ideal evaluation metric due to its emphasis on executable code. However, our strategy prioritizes extensive data utilization to accurately reflect the capabilities of models across thousands of codebases. Implementing pass@k involves ensuring the executability of each piece of code, which is a complex task given the diverse environmental configurations and dependency issues common in library code. The logistical challenges associated with verifying the executability of such a vast array of code samples render pass@k less feasible for our study's scope. Nonetheless, we agree that pass@k could be a valuable addition in a more controlled or smaller-scale setting and will consider it for future research where applicable.
> >
> > As for CodeBLEU, we appreciate your suggestion and have taken steps to incorporate it into our evaluation. Our latest experiments, as detailed in the Global Response, already include CodeBLEU as an additional metric. This inclusion should provide a more rounded evaluation of the models’ capabilities.
> >
> > Despite these challenges, we believe that EM, ES and CodeBLEU are effective in reflecting the performance of models on dataset with a large scale.
> >
> > > Expanding Model Selection for RepoBench-C Evaluation
> >
> > Following your advice, we have conducted a test using the 7B version of SoTaNa. The results, which are detailed in the global response section, indicate that SoTaNa, despite being relevant, has limitations in code completion. We suspect these limitations are due to its foundation on the Llama model, which is not primarily a code model, and its fine-tuning on a context length of 512, not ideal for our settings of "long context" evaluation. Nonetheless, SoTaNa is definitely a relevant work and adds valuable insights to our study. Regarding the incorporation of more diverse Language Model Models (LLMs), our focus was primarily on base models. However, in light of your feedback, we have decided to broaden our scope by including CodeLlama in its various configurations (7B, 13B, 34B). Please check the added results in global response, where shows the impressive performance of CodeLlama.

---

### Official Review · Reviewer_zCY2 · 2023-10-31

**Soundness:** 3 good
**Presentation:** 4 excellent
**Contribution:** 3 good
**Rating:** 8
**Confidence:** 4

**Summary:**

The authors address a need for repository wide benchmarks for code-prediction and code-retrieval tasks. They do so by creating two datasets, in the test set, they recover repository information for the github-code dataset and create two variants, a 2K and an 8K variant. For the test set, they crawl permissively licensed Java and Python projects after the The Stack cut-off date. To better mimic real-world scenarios, the test set is not separated by prompt length. As for the benchmark itself, it focuses on three tasks that should exercise both cross-file and in-file context requirements. The tasks are code auto-completion, code-retrieval, and a join task where the relevant cross-file information should be retrieved before it is used for completion (pipeline).

**Strengths:**

(1) The paper addresses the need for a repository wide benchmark that better aligns with real-world usecases in software projects. (2) It addresses data leakage issues* by crawling new data for the test set and (3) provides fine-tuning data for models that may require it.
(4) The StarCoder overfitting to file-level use-cases provides interesting additional insight.

**Weaknesses:**

The main concerns with the paper are two-fold.

The usefulness of the benchmark relies on a gentleman agreement to not use data from the collection dates during training or fine-tunning.

Another concern is the opt-out possibility. While not strictly necessary, a nice-to-have would be an opt-out mechanism similar to the The Stack one for authors that may want to remove their code from.the data.

**Questions:**

Is there an intention to make the bechmark a "living" benchmark where the test set is periodically refreshed to be past the training set horizon date?

Alternatively, is there an intention to check and disqualify models that have trained on test set data?

**Details Of Ethics Concerns:**

The opt-out is more-so a nice-to-have rather than necessary since the authors have taken care to respect code licenses during crawling.

---

> ### Author Response · Authors · 2023-11-22
> **Response to Reviewer zCY2**
>
> Dear Reviewer zCY2,
>
> Thank you for your appreciation and constructive feedback on our paper.
>
> > Opt-Out Possibility and Living Benchmark
>
> We are grateful for your suggestion about the opt-out mechanism and the idea of a living benchmark. Interestingly, we are already moving in this direction and the living benchmark is exactly what we want to present since we have the pipeline for construction and we can always crawl the newest data for benchmarking.
>
> Specifically, we are collaborating with the StarCoder team, and we will present the newest version of RepoBench shortly with all the repositories are created in September, 2023, aligning with the next release of "The Stack." As for the opt-out mechanism, the StarCoder team has implemented a rigorous deduplication process on their new training data, which will also be open-sourced soon.
>
> > Alternatively, is there an intention to check and disqualify models that have trained on test set data?
>
> Thanks for pointing out this. RepoBench as a living benchmark as you expected, it allows us the flexibility to continuously update and release new versions (e.g., September version, October version, November version, etc.). So, if a model has been trained and released, we can simply validate it against the latest version of RepoBench. This method ensures an effective check against any potential training on test set data, as models will be evaluated on newly updated benchmarks that they have not encountered before, ensuring the integrity and fairness of the evaluation process.

---

> > ### Comment · Reviewer_zCY2 · 2023-11-22
> >
> > Thank you for the answers and confirming the move to a living benchmark as well as the global preliminary results response above.
> >
> > If the opt-out mechanism is shared with "The Stack", I am more than satisfied with the process as last I tested it, it was fairly smooth and simple to follow.
> >
> > As for data leakage, the cat-and-mouse game remains, but the move to a living benchmark alleviates my concerns as new cuts of the benchmark should reveal models that may have cheated on previous iterations.

---

> > > ### Author Response · Authors · 2023-11-22
> > > **Opt-out mechanism**
> > >
> > > Thank you for your insightful feedback and support.
> > >
> > > We are also planning a opt-out process akin to 'The Stack' for our subsequent versions of releases (e.g., Oct 2023 Version). This should involve crawling the newest data, checking licenses, and then launching an `Am I in RepoBench` API, similar to [Am I In the Stack](https://huggingface.co/spaces/bigcode/in-the-stack), to enhance data inclusion transparency. We will allow a designated period, for opt-outs. After this, we will proceed to remove any opted-out data before constructing the test benchmark.
> > >
> > > Thank you once again for your suggestions.

---

### Official Review · Reviewer_V3UP · 2023-10-31

**Soundness:** 3 good
**Presentation:** 3 good
**Contribution:** 2 fair
**Rating:** 6
**Confidence:** 3

**Summary:**

Authors propose RepoBench - a benchmark for repository level code auto-completion evaluation. They propose three evaluation tasks: retrieval, code completion, and pipeline. Authors perform experiments using RepoBench

**Strengths:**

- Significant work on RepoBench construction.
- Extensive experiments with RepoBench with existing models and retrieval techniques.

**Weaknesses:**

- It is not clear what new insights RepoBench and experiments on it contribute to the field. Were the results previously unknown or unexpected?

- This might not be a weakness of the paper per se, but it concerns me a bit that random retrieval is close to or even outperforms some non-random retrieval methods.


I increased the rating based on authors' answer to my questions.

**Questions:**

- What is exactly "the first appearance of a cross-file line within a file"? Is this the import line? Is this the first line that uses cross-file function?

---

> ### Author Response · Authors · 2023-11-22
> **Response to Reviewer V3UP**
>
> Thank you for your thoughtful review and valuable comments on our paper. We appreciate the opportunity to clarify and elaborate on the aspects you've highlighted.
>
> > It is not clear what new insights RepoBench and experiments on it contribute to the field. Were the results previously unknown or unexpected?
>
> We understand your concern regarding the novel insights provided by RepoBench. Our primary contribution through RepoBench is in addressing the need for benchmarks that evaluate repository-level code auto-completion, a gap that exists in current benchmarks focused mostly on single-file tasks. This focus is particularly relevant given that real-world applications of code models often require a comprehensive repository-level approach for optimal functionality.
>
> Additionally, RepoBench has the ability to construct generation tasks with customizable context lengths (e.g., 2k, 8k, 16k, 64k, etc.). This flexibility allows for an in-depth analysis of model performance under varying long-context scenarios
>
>
> During our experiments, we did encounter several intriguing findings that were unexpected and might be insightful:
>
> 1. **Multilingual Code Generation**: While current models show promising results in Python, matching Codex in some cases, we observed a noticeable performance gap in Java code generation
> 2. **Long Context Generation**: On repobench-c with 8k max length, StarCoder is still trailing behind the capabilities of Codex, highlighting a crucial area for future development.
> 3. **Repo-Level performance**: Our initial assumption was that repo-level code generation would be generalizable. However, we found that models like StarCoder did not perform well at this level, suggesting they might need to be trained with repository-level data.  This insight led to a collaborative effort with the StarCoder team and our subsequent contribution to the model improvement. We show some of the preliminary ablation study in a global response, which demonstrates that repository-level training can improve the performance dramatically.
> 4. **Retrieve as much as possible**: Merely retrieving the gold code snippet is not always the most effective approach. Instead, filling the context with as much 'relevant' code as possible from the repository tends to improve performance.
> 5. **Positioning of Cross-File Context**: Unlike long context generation in textual documents where models pay more attention to the beginning of the prompt, in repository-level code generation we found that placing more relevant cross-file snippets towards the end of the prompt (closer to the in-file context) can improve performance.
>
> > It concerns me a bit that random retrieval is close to or even outperforms some non-random retrieval methods.
>
> The observation regarding random retrieval occasionally outperforming strategized retrieval, particularly in Java, is indeed intriguing. We speculate that this phenomenon is influenced by the inherent characteristics of the language. In Java, cross-file references often involve class definitions (e.g., `public class DerivedClass extends BaseClass {`). In this scenario, previous lines might not provide contextual clues (meaningful information) for retrieval. While in Python, cross-file references may often involve more direct contextual clues such as function calls with parameters, making strategized retrieval more effective. For example, if you try to use `calculate(left, right)`, you may define `left=?` and `right=?` in the previous lines. Thus, our findings suggest that effective retrieval strategies may vary between different programming languages.
>
> > What is exactly "the first appearance of a cross-file line within a file"? Is this the import line? Is this the first line that uses cross-file function?
>
> We acknowledge for the ambiguity in our description. The "first appearance of a cross-file line within a file" refers to the first instance where a cross-file function/class is used in the code, not the import line. So strictly speaking, it should be the second appearance (the first one will always be the import line).

---

> > ### Comment · Reviewer_V3UP · 2023-11-23
> > **Thanks to authors for the answers**
> >
> > Thank you for responses and elaboration on your work. I have increased the rating of the paper.

---

### Author Response · Authors · 2023-11-22
**Global Response (1/2)**

We are grateful to all reviewers for their detailed and constructive feedback. We are encouraged that reviewers see value in our work on repository-level code retrieval and completion benchmarks. In particular, reviewers highlighted:

1. RepoBench addresses the need for repository-wide benchmarks relevant to real-world software projects. (`zCY2` and `mVr1`)
2. The effort in creating RepoBench and the rigor of our experiments. (`V3UP`)
3. Avoiding data leakage by using newly crawled data for the test set. (`zCY2`)
4. Interesting additional insights into the challenges and nuances of repository-level code completion. (`zCY2`)
5. Well-written and easy to follow (`mVr1`)

Each constructive comment from the reviewers was deeply deliberated upon, and we have endeavored to address the pivotal concerns that have been raised. In global response, we will mainly present our latest research results and added experiments.

### RepoBench is a living benchmark

- As suggested by Reviewer zCY2, we are indeed making RepoBench a "living" benchmark. Our data construction pipeline allows for the continuous update of data, ensuring that the benchmark remains not-data-leakage and challenging.
- We will release RepoBench's September 2023 version soon. This update will also include deduplication based on the next version of The Stack dataset.

### Repo-level pretraining imrpoves repo-level code completion.

- Our collaboration with the StarCoder team has yielded preliminary ablation studies, illustrating the impact of repo-level pretraining.
- They provided some 1B models, trained on the same tokens of data (from the upcoming release of The Stack), with an 8k context length.
- The training data varied between repo-level and file-level, with different strategies for retrieving repo-level code (random and depth-first) to construct data point for repo-level.
- As shown in Table 1 below, our results demonstrate a clear advantage in using repo-level pretraining for repo-level code generation.

Table 1: Ablation studies of 1b models using repo-level and file-level data for training.

| Training Level | Cross-File Context | EM (XF-F) | ES (XF-F) | EM (XF-R) | ES (XF-R) | EM (IF) | ES (IF) | EM (All) | ES (All) |
| -------------- |--------| --------- | --------- | --------- | --------- | ------- | ------- | ------- | ------- |
|File | ❌| 10.72 | 54.54 | 32.53 | 67.66 | 26.97 |63.98 | 23.32 |62.01 |
| | ✅ | 13.38 | 54.96 | 34.58 | 68.93| 28.92| 64.98 | 25.55 | 62.90 |
|Repo (random) | ❌|10.99 | 55.28 | 32.94 | 69.63 | 26.57 | 64.92 | 23.42 | 63.22 |
| | ✅ | 14.29 | 56.69 | 35.10 | 70.09 | 29.51 | 65.75 | 26.22 | 64.12 |
|Repo (depth first) | ❌| 10.72 | 55.00 | 32.26 | 69.10 | 27.08 | 64.67 | 23.27 | 62.87 |
| | ✅ | 14.82 | 57.30 | 35.13 | 69.67 | 29.62 | 65.56 | 26.45 | 64.13 |

### CodeLlama Results on RepoBench-2k and 8k with CodeBLEU score included.

Additionally, we have included results from the CodeLlama model series on RepoBench-C 2k and 8k datasets, incorporating the CodeBLEU score (as suggested by Reviewer mVr1) for a comprehensive evaluation:

For a clearer and more concise presentation, we choose not to display the separate categories of XF-F, XF-R, and IF in our results. We only present the average weighted results across these settings.

Table 2: Results on RepoBench-C (2k).

| Language| Model                     | Params. | Exact Match | Edit Sim | CodeBLEU |
|-------|---------------------------|---------|-----------|-----------|-----------|
| **Python** | CodeGen                  | 6.1B   | 31.67 | 70.67 | 42.15 |
|       | SoNaTa | 7B |  13.95 | 58.97 | 26.79 |
|       | CodeLlama | 7B | 34.11 | 71.24 | 43.46 |
|       | CodeLlama | 13B | 36.18 | 72.25 | 45.60 |
| |StarCoder                | 15.5B  |  31.67 | 71.27 | 41.46 |
|       | CodeGen                  | 16.1B  | 33.41 | 71.20 | 43.58 |
|       | CodeLlama | 34B | **37.40** | **72.98** | **47.04** |
|       | Codex                    | -   | 31.31 | 72.21 | 41.45 |
| **Java**   | CodeGen                 |  6.1B   |  29.59 | 70.27 | 44.86 |
|       | SoNaTa | 7B |21.1 | 66.21 | 37.15 |
|       | CodeLlama | 7B | 35.88 | 76.75 | 49.98 |
|       | CodeLlama | 13B | 37.96 | 77.77 | 52.00 |
|       | StarCoder                | 15.5B  |  37.35 | 77.00 | 51.81 |
|       | CodeGen                  | 16.1B  |  30.45 | 70.29 | 45.94 |
|       | CodeLlama | 34B | 39.41 | 78.52 | 53.56 |
|       | Codex                    | -  | **42.47** | **80.01** | **55.61** |

---

> ### Author Response · Authors · 2023-11-22
> **Global Response (2/2)**
>
> Table 3: Results on RepoBench-C (8k).
>
> | Language| Model                     | Params. | Exact Match | Edit Sim | CodeBLEU |
> |-------|---------------------------|---------|-----------|-----------|-----------|
> | **Python** | CodeLlama                  | 7B   | 33.24 | 70.44 | 43.14 |
> | | CodeLlama                  | 13B   | 35.56 | 71.57 | 45.10 |
> | | StarCoder                  | 15.5B   | 26.84 | 68.39 | 37.71 |
> | | CodeLlama | 34B | **36.26** | **72.19** | **45.71** |
> | | Codex | - | 32.13 | 71.89 | 42.27 |
> | **Java** | CodeLlama                  | 7B   | 33.45 | 74.33 | 47.64 |
> | | CodeLlama                  | 13B   | 36.26 | 75.72 | 49.44 |
> | | StarCoder                  | 15.5B   |28.45 | 66.31 | 44.94 |
> | | CodeLlama | 34B |36.84 | 76.06 | 50.77 |
> | | Codex | - | **40.52** | **77.97** | **53.63** |

---

### Meta-Review · Area_Chair_cKXN · 2023-12-08

**Metareview:**

The paper introduces a repository level benchmark for code generation, with evaluations of retrieval (R), code completion (C), and end-to-end (Pipeline, P = R+C). The authors made a serious effort to benchmark several public models, although they omitted models that have only API access (e.g. GPT4). Repository level benchmarks are very relevant for evaluating beyond function or class level (as most code generation benchmarks do for now), and indeed (for instance) SWE-Bench (Jimenez et al. 2023) was made public right after the ICLR submission deadline. While it is difficult to predict benchmark popularity at this step, I believe this paper and benchmark constitutes a useful contribution to the community. It has some quality control (deduplication, rebuildability), splitting the end-to-end task in 2 smaller evaluations makes it more approachable and gives natural ablations/insights. Three reviewers recommend acceptance, one recommend rejection, but they did not answer the authors rebuttal which seems to address several of their concerns. Overall, I recommend to accept this paper for publication at ICLR.

**Justification For Why Not Higher Score:**

N/A

**Justification For Why Not Lower Score:**

There is a flurry of new benchmarks in code generation coming, and there is a very similar SweBench, but I think this is a valuable addition to benchmarks, the process to build and rebuild this benchmark may make it stand the test of time. The authors made the effort of launching the benchmark with also a lot of popular models benchmarked.

---

### Decision · Program_Chairs · 2024-01-16

Accept (poster)